# Imputing Biomarker Status from RWE Datasets—A Comparative Study

**DOI:** 10.3390/jpm11121356

**Published:** 2021-12-13

**Authors:** Carlos Traynor, Tarjinder Sahota, Helen Tomkinson, Ignacio Gonzalez-Garcia, Neil Evans, Michael Chappell

**Affiliations:** 1School of Engineering, University of Warwick, Coventry CV4 7AL, UK; Neil.Evans@warwick.ac.uk (N.E.); m.j.chappell@warwick.ac.uk (M.C.); 2Clinical Pharmacology and Quantitative Pharmacology, Clinical Pharmacology and Safety Sciences, AstraZeneca, Cambridge CB2 1RY, UK; tarjinder.z.sahota@gsk.com (T.S.); helen.tomkinson@astrazeneca.com (H.T.); ignacio.gonzalez@astrazeneca.com (I.G.-G.)

**Keywords:** imputation, real-world evidence, survival, simulation, machine-learning, statistical inference

## Abstract

Missing data is a universal problem in analysing Real-World Evidence (RWE) datasets. In RWE datasets, there is a need to understand which features best correlate with clinical outcomes. In this context, the missing status of several biomarkers may appear as gaps in the dataset that hide meaningful values for analysis. Imputation methods are general strategies that replace missing values with plausible values. Using the Flatiron NSCLC dataset, including more than 35,000 subjects, we compare the imputation performance of six such methods on missing data: predictive mean matching, expectation-maximisation, factorial analysis, random forest, generative adversarial networks and multivariate imputations with tabular networks. We also conduct extensive synthetic data experiments with structural causal models. Statistical learning from incomplete datasets should select an appropriate imputation algorithm accounting for the nature of missingness, the impact of missing data, and the distribution shift induced by the imputation algorithm. For our synthetic data experiments, tabular networks had the best overall performance. Methods using neural networks are promising for complex datasets with non-linearities. However, conventional methods such as predictive mean matching work well for the Flatiron NSCLC biomarker dataset.

## 1. Introduction

Analysing Real-World Evidence (RWE) datasets in clinical research have increasing utility in overcoming some limitations of conventional Randomised Clinical Trials (RCT), such as target sourcing and early patient stratification. One of the biggest challenges when working with RWE datasets is that experimentation is limited, and variables are often incomplete. Missing data appear as gaps in the data set that hide meaningful values for analysis. As Little put it, “the best resolution for handling missing data is not to have missing data” [1]. However, analysing RWE datasets poses the challenge of handling missing values. Indeed, missing data are found not only in observational RWE datasets but also in RCT [2]. Using RWE effectively to advance current medical practice requires finding better solutions to the missing value problem.

In RWE datasets, there is a need to understand which biomarkers best correlate with clinical outcomes to facilitate drug development. A substantial difficulty in real-world settings is that several biomarkers statuses may be missing in the dataset, hiding meaningful information in the analysis. Hence, excluding the underlying value of missing data may invalidate the results. There are many practical implications when missing data are present; for example, it can lower the power, affect the precision of the confidence intervals for parameter estimates, and lead to biased estimates.

We study the scenario where the observed features, sometimes called covariates of interest, are possibly incomplete. Let us denote the features by *X*. The missingness mechanism $R^{X}$ places the missing values in $X^{obs}$ masking the actual value *X*, i.e., $R^{X}$ is a random variable taking values in $\left\{0,1\right\}$, so that:$$X_{i}^{obs}=\left\{\begin{array}{cc} X_{i} & \mathrm{if}\;R_{i}^{X}=0\\ n.a. & \mathrm{if}\;R_{i}^{X}=1 \end{array}\right.$$

Rubin et al. [3] defined three possible scenarios for missing data: Missingness Complete at Random (MCAR), Missingness at Random (MAR), and Missingness Not at Random (MNAR). All types of Missingness can be classified usefully into this taxonomy, which indicates the most appropriate procedure to minimise bias. We say that the data are MCAR if the complete dataset *X* do not influence the mask $R^{X}$, i.e., *X* are independent of $R^{X}$. In large sample theory, we may test for MCAR from the data by conducting Little’s test [4], which compares the change in the empirical mean of measured variables $X^{obs}$ if removing cases with missing values. In MAR, other measured variables are influencing the missingness mechanism. The consequences of MAR are diverse but addressable by controlling for other known variables and imputation. In analysing RWE datasets, it is helpful to assume MAR because it allows handling missing data with general-purpose imputation algorithms, the central topic of this article. The MNAR scenario is complicated, and if present, it means that $X^{obs}$ may induce a statistical bias when estimating parameters with missing values. It is also impossible to test whether X influences its missingness mechanism, $X^{obs}$, with the given data [5]. Possible resolutions are to model the MNAR mechanism, collect more data on the missing variable, or collect information on other variables that allow modelling MNAR as a MAR scenario.

A conventional ad-hoc method to handle missing data is the complete case analysis, to delete any rows or columns with missing variables. The problem with complete case analysis is that it squanders information reducing the sample size considerably, especially in RWE scenarios where incomplete cases may be frequent. Imputation algorithms are general strategies that replace missing values (n.a.) with plausible values. Nevertheless, replacing missing values with single static values cannot be completely representative of the missing sample. After all, imputed values are estimated, not observed. Therefore, it is often more appropriate to apply a random variable approach to represent missing values. Rubin et al. [6] proposed multiple imputations (MI) for survey non-responders to tackle the uncertainty in the missing values that single imputation cannot represent with a point estimate. The general idea of MI is to generate multiple complete datasets, analyse each dataset separately, and summarise the results. Imputation algorithms that perform MI must replace missing values with samples from the missing values’ joint probability density function. Therefore, the MI approach embraces the uncertainty in the missing values that single imputation with a point estimate cannot represent. Early papers proposing imputation algorithms for MI often apply conventional statistical methods to estimate the probability density function like expectation-maximisation (EM) [7]. More sophisticated methods, adapting ideas from Markov-Chain Monte-Carlo [8], dimension reduction [9], ensemble learning [9] and deep learning [10], have been proposed. However, there is a lack of literature on validation, and systematic comparison of imputation methods [11]. Furthermore, the importance of considering missingness patterns and the data distribution when comparing methods has received little attention [12]. Neglecting to do this may lead to biased results concerning the relative performance of imputation methods.

The present paper develops a new multivariate imputation algorithm powered by a deep neural network (DNN) for tabular data named Tabnet [13] that performs multiple imputations and supports mixed data types. We define the causal mechanism of missing data and explain the rationale for considering imputation algorithms in RWE data examples. Following best practices in developing a new algorithm for imputation [3], we aim to find an accurate imputation algorithm that provides good statistical properties, such as unbiased parameter estimates and coverage of the parameter estimates determined from sampling and missing data variance. The RWE data source used in the present paper was the Non-Small Cell Lung Cancer (NSCLC) Flatiron database [14], a dataset of de-identified patient-level electronic medical records in the United States spanning 280 community practices seven sizeable academic research institutions. The Flatiron NSCLC biomarker cohort potentially differs from other dense sampled datasets, where biomarkers are measured longitudinally. However, it is a typical example of the use case of RWE, where clinical interest is often on biomarkers tested at cancer diagnosis that help identify sub-populations that most benefit from targeted treatments. For NSCLC, clinical practice guidelines recommendations include testing the genomic biomarkers Epidermal growth factor receptor (EGFR), Anaplastic lymphoma kinase (ALK), Kirsten rat sarcoma (KRAS), B-RAF proto-oncogene (BRAF), and immunotherapy marker programmed death-ligand 1 receptor (PDL1).

The present article makes several contributions that we summarise as follows:We propose a systematic approach for comparison of imputation methods on RWE datasets.We apply this approach to compare the model performance of seven imputation methods. The six methods are expectation-maximisation (EM), predictive mean matching (PMM) with multivariate imputations by chained equations (MICE), bootstrap-based principal component analysis (MIPCA), one method that uses random forest (MIRF), generative adversarial imputation networks (GAIN) and a method that uses mice with tabular networks (MITABNET).We conduct a comparative study of the state-of-the-art imputation algorithms in simulations and RWE data benchmarks with clinical oncology applications.Our research develops a new multivariate imputation algorithm powered by a deep neural network (DNN) for tabular data named Tabnet [13] that performs multiple imputations and supports mixed data types.

## 2. Materials and Methods

### 2.1. RWE Dataset Analysed

The dataset analysed consists of patients who received a diagnosis of advanced NSCLC. The inclusion criteria are patients aged ≥18, pathological confirmation of NSCLC obtained from tumour cytology or biopsy, documented diagnosis of unresectable TVM Stage III-IV NSCLC, and at least one biomarker status of EGFR, ALK, KRAS, BRAF, PDL1. The dataset analysed includes 35,012 individuals, see Table 1.

This paper analyses the impact of biomarker status ALK, BRAF, EGFR, KRAS, and PDL1 on real-world survival analysis. Even though one can use unknown status as a predictive marker in a multivariate survival model, we argue that such an analysis would not be helpful for clinicians seeking to understand the impact of biomarker status on clinical outcomes. Figure 1 displays the combinations of missingness for the biomarkers EGFR, ALK, KRAS, BRAF, PDL1 in the RWE dataset, with a missingness mechanism that Little’s test [4] suggests is not MCAR (p<0.0001).

### 2.2. Methods for Multiple Imputations

Different approaches to drawing multiple imputations exist in the literature. The following is a brief description of the battery of algorithms compared in the present paper.

#### 2.2.1. Expectation-Maximisation

Honaker et al. [7] built on the expectation maximisation (EM) approach to impute missing values and performed multiple imputations using a bootstrap-based design. The EM algorithm iteratively starts with an expectation step that calculates the likelihood function given by the expected complete data conditional on current parameter estimates. The expectation step is hence, a form of imputation. Then, the maximisation step chooses the model parameters by optimising the likelihood function. For a detailed explanation of the EM imputation algorithm, we refer to [7]. We used the implementation of EM in the R package Amelia [7].

#### 2.2.2. Predictive Mean Matching

Gerko et al. [16] suggest a Gibbs sampler for the multivariate imputations by chained equations (MICE) algorithm to draw from an approximate posterior distribution after evaluating the complete data replacing missing values for placeholders. MICE is an iterative method to use regression strategies such as predictive mean matching (PMM) for any variable. For a detailed description of the MICE-PMM algorithm, we refer to [16]. We use the stable R released package mice for PMM [8].

#### 2.2.3. Random Forest

Stekhoven et al. [9] suggested a random forest algorithm for missing data imputation that can perform multiple imputations (MIRF) by running the algorithm with different random seeds. The missing random forest algorithm issues predictions for missing values by weighing many relatively uncorrelated trees. Ref. [11] implements the random forest algorithm iteratively by fitting the observed values and updating the missing values until meeting a model performance stopping criterion. We use the implementation of missing random forest in the R package missForest [9].

#### 2.2.4. Factorial Analysis

Josse et al. [17] proposed factorial analysis of mixed data (FAMD) methods that exploit the global similarity between individuals and the correlation between variables to impute missing datasets. The FAMD algorithm starts calculating the FAMD components and then projects each principal component using the FAMD prediction. Iterative FAMD repeats these steps until convergence. The FAMD algorithm is flexible enough to perform multiple imputations via a non-parametric bootstrap. We estimate the number of components for FAMD using cross-validation. For a detailed explanation of the regularised iterative FAMD algorithm, we refer to [17]. We use the regularised iterative FAMD in the R package missMDA [17].

#### 2.2.5. Generative Adversarial Imputation Networks

Yoon et al. [10] adopted the generative adversarial framework from [18] to develop a Generative Adversarial Imputation Network (GAIN), which is a non-stochastic neural network constituted by a generator and a discriminator network trained in a zero-sum game. The discriminator attempts to distinguish the imputed values from the actual ones, which predicts the mask. The generator, on the other hand, attempts to deceive the discriminator by generating more realistic imputations. The generator inputs are the mask and the original data with missing values that are substituted by noise, e.g., from a Normal random variable. We use the Python implementation of GAIN [10].

#### 2.2.6. Multiple Impuations with Tabnet

The multiple imputations with tabular nets (MITABNET) imputation algorithm, given by Algorithm 1, build on that developed in Tabnet [13] for instance-wise and non-linear covariate effects modelling. For single imputation, the input features include, by default, all variables except the one with missing values. Mimicking ensembling Tabnet uses sequential steps. Each step starts with a Feature Transformer Block, followed by an Attentive Transformer Block that creates the mask, followed by another Feature Transformer Block. The output makes both predictions and the input for the next step. For details on the architecture of Tabnet, we refer to [13].

The algorithm’s predictions for missing values, i.e., single imputations, are the sum of the step’s outputs, passed to a final fully connected layer to resolve any regression or classification problems. The cost function of Tabnet is given by:$$J\left(\theta \right)=\left\{\begin{array}{c} {\left(x^{\prime }-x\right)}^{2},\mathrm{if}\;x\;\mathrm{is}\;\mathrm{continuous}\\ -\sum_{k}^{K}x_{k}\log\left(x_{k}^{\prime }\right),\mathrm{if}\;x\;\mathrm{is}\;\mathrm{categorical} \end{array}\right.$$
where θ denotes the network’s parameters, for continuous variables $J\left(\theta \right)$ is defined as the mean squared error and for binary and categorical variables as the cross-entropy. We use validation error by splitting the available data into train and validation datasets (80:20). The optimisation algorithm that we use is the Adams optimiser [19], an extension to stochastic gradient descent, with decay learning rate (step size = 10, γ=0.9).

We apply multivariate imputations by chained equations to perform multiple imputations with Tabnet sequentially, starting with random draws from the observed data $X_{j}^{obs}$. The MITABNET algorithm, see the pseudo-code in Algortihm 1, is repeated for each m=1, ...,M imputed datasets as follows:Initialise the complete dataset with random samples from the observed dataset $X^{obs}$.For each variable, split the missing and observed datasets, split the observations into training and validation sets (80:20), use Tabnet to learn the distribution $P(X_{j}|X_{-j},R,\theta )$.Use Tabnet to learn the target distribution. The approach then uses the trained Tabnet to predict the missing values in $X_{j}$.

Step 1 to 3 are repeated a prespecified number of iterations for each *K* feature with missing values. We implement MITABNET as an hybrid R and Python package and provide the implementation in SM.
  **Algorithm 1:** MITABNET’s pseudo-code.  
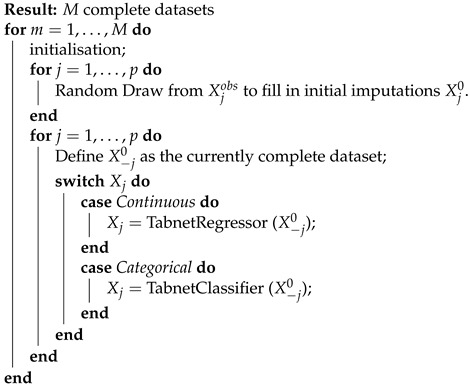


### 2.3. Strategy for Comparing Methods

A head-to-head comparison of imputation methods involves the ability of the algorithms to recover the actual value from an “amputated” dataset. By amputation, we refer to the concept developed in [12], where a simulation algorithm generates the missingness mechanism to obtain datasets that have missing values following a specific pattern. The following sub-section demonstrates a systematic approach to comparing imputation algorithms on RWE datasets.

#### 2.3.1. Analysis of Interest

The analysis of interest is survival analysis. The survival analysis had the following relevant parameters: index date and the end date. We define the index date as the start date of treatment anchoring the survival analysis. We define the end date as the death date for patients for whom this is known or the last confirmed activity for patients for whom it is not known. The time at risk is the difference between the end date and the index date. We used Effron’s likelihood [20] to handle tied death times as implemented in the **rms** R package [21]. To be consistent with all imputation methods, we use the same multivariate Cox proportional hazards model [22], given by:(1)$$h_{i}\left(t\right)=h_{0}\left(t\right)\exp\left[\mu_{i}\right]$$
where $\mu_{i}$ is the prognostic index for the ith individual, given by:(2)$$\mu_{i}=\beta_{\mathrm{ALK}}\cdot {\mathrm{ALK}}_{i}+\beta_{\mathrm{BRAF}}\cdot {\mathrm{BRAF}}_{i}+\beta_{\mathrm{EGFR}}\cdot {\mathrm{EGFR}}_{i}+\beta_{\mathrm{KRAS}}\cdot {\mathrm{KRAS}}_{i}+\beta_{\mathrm{PDL}1}\cdot {\mathrm{PDL}1}_{i}$$
where $\beta_{P}$ are the log-hazard ratios of each *P* biomarker and ALK, BRAF, EGFR, KRAS and PDL1 are indicator variables. The pairs plot matrix in Figure 2 shows the kernel density estimates of the prognostic index ($\mu_{i}$) in the Flatiron NSCLC data for each imputation method with pairwise scatter plots calculated on the off-diagonal. The pairs plot shows that the different imputation methods’ μ are positively correlated but not collinear, indicating that there may be practical differences in the post-imputation prediction performance.

To combine inference in the frequentist Cox proportional hazard model, Equation (2) above, we calculate a parameter estimate for each imputed dataset ($\beta_{m}$) and use Rubin’s rule [24] to average over the estimates. The formula for the point estimate of each parameter estimate ($\bar{\beta }$) is an average over the point estimates of each imputed dataset, such that:(3)$$\bar{\beta }=\frac{1}{M}\left(\sum_{m=1}^{M}\beta_{m}\right)$$

Furthermore, multiple imputations allow us to compute the total variance, which is helpful for constructing confidence intervals for the parameter estimates. To do so, we must consider the sampling variability within each complete dataset, denoted as $\sigma_{w}$, and related to the conventional estimation of variance, which is given by:(4)$$\sigma_{w}=\frac{1}{M}\left(\sum_{m=1}^{M}\sigma_{m}\right)$$

Besides, we must consider the variability between complete datasets, denoted as $\sigma_{b}$, regarded as being the result of missing values, such that:(5)$$\sigma_{b}=\sqrt{\frac{\sum_{m=1}^{M}{\left(\beta_{m}-\bar{\beta }\right)}^{2}}{M-1}}$$

Finally, to compute the total variance of the parameter estimates, we apply the formula adapted from [25] by combining $\sigma_{w}$ and $\sigma_{b}$, such that:(6)$$\sigma =\sigma_{w}+\left(1+\frac{1}{M}\right)\sigma_{b}$$

Using the adjusted formula to calculate the degrees of freedom [24], it is straightforward to compute a confidence interval for $\bar{\beta }$ with α value, given by:(7)$$\bar{\beta }\pm t_{\mathrm{df},\frac{1-\alpha }{2}}\sqrt{\sigma }$$

#### 2.3.2. Single Imputation Accuracy

To evaluate the imputation accuracy for single imputations, we report the root mean squared error, given by:(8)$$\sqrt{\sum_{i=1}^{N}\frac{{\left(x_{i}^{\prime }-x_{i}\right)}^{2}}{N}}$$
where $x_{i}^{\prime }$ is the actual amputated value for the *i*th individual’s *x* variable, and $x_{i}$ is the imputed value.

#### 2.3.3. Performance of the Multiple Imputation Algorithm

Accuracy use is a convenient yardstick for benchmarking imputation algorithms [10]. However, for inference, our interest is in the distributional characteristics of the multiply imputed datasets, such as preserving parameter estimates’ moments having low bias, high coverage of confidence intervals, and the robustness to missingness mechanisms such as MAR. Let a survival dataset given by *Y* the outcome space comprised by the survival time and the censoring indicator, and $X_{p}$ the feature space with 1, ...,p features.

Once we compute $\bar{\beta }$ and σ for each parameter estimate, we evaluate the imputation performance of each imputation algorithm, by evaluating the following heuristics:Percentage Bias: An optimal imputation algorithm should be unbiased. We compute the parameter estimate $\hat{\beta }$ from the complete dataset before running the amputation algorithm, see Equation (Equation 10). For each parameter, the percentage bias is given by $100\cdot \left|\frac{\sum_{p=1}^{P}\left({\bar{\beta }}_{p}-{\hat{\beta }}_{p}\right)}{\sum_{p=1}^{P}\left({\hat{\beta }}_{p}\right)}\right|$.Width of Confidence Intervals: With σ computed with Equation (Equation 6) we compute 95% Wald confidence intervals (CI). A narrow confidence interval that covers the parameter of interest $\hat{\beta }$ are prefered. However, smaller confidence intervals that cover the parameter interest $\hat{\beta }$ indicate sharper inference.Coverage: With a 95% (CI), we compute if it includes the original parameter of interest $\hat{\beta }$. We repeat this computation 200 times and compute the proportion of times that $\hat{\beta }$ is inside the  95% CI.Convergence: To monitor convergence, we use the trace-plot and the $\hat{R}$ metric [26]. In general, the $\hat{R}$ metric evaluates how well Markov Chains mixed, within and between chains. Therefore, we suggest it is a helpful statistic for imputation algorithms that rely on the Gibbs sampler.

#### 2.3.4. Generation of Sample Datasets for Multiple Comparisons

We will not know the resulting criteria for the comparison study based on the data distribution from only the available complete data because of the severity of the missing values in RWE data sets. Instead, one can impute the original data with each of the imputation algorithms (Home imputation), which will generate a concept drift that we define as the shift in the data distribution induced by the imputation method. Then, one can sample amputated datasets with a MAR mechanism similar to the original data by using the original mask matrix given by the indicators of the cells that were missing in the original data, see Figure 3. We suggest using a Bernouilli random variable since is the maximum entropy distribution for binary events. We set the probability of missingness *p* to 0.25 for the initially observed cells and 1−p=0.75 for the initially missing cells. The value of p=0.25 strikes a balance in bootstrapping the missingness pattern from the original data set while conducting a fair head-to-head comparison among the imputation algorithms (Visiting imputation). Note that setting *p* to 0.5 would yield an MCAR mechanism of missingness. The method has been used for benchmarking imputation algorithms [9]. To illustrate the dependence on the concept drift, we evaluate the imputation performance using three imputation methods on amputated data sets samples of the FlatIron NSCLC cancer biomarker data. Table 2 shows the percentage bias results for two such samples. Let us look at the first re-imputed sample. The percentage bias varies depending on the imputation algorithm used to obtain the complete data set. Moreover, EM obtains the lowest percentage bias for the data set imputed originally with MITABNET, which contrasts with the results from the amputated sample 2, where MITABNET obtained the lowest bias for the data set imputed originally with MITABNET.

The inconsistent results from samples 1 and 2 illustrate that one needs several samples to evaluate the performance of the imputation methods. Another potential weakness of investigating only one sample is that if the imputation algorithm to generate the dataset is wrong, the competing algorithms might appear to be incorrectly biased. Therefore, we draw many samples *S* amputated datasets, yielding *S* different values for each criterion. We draw *S* independent amputated datasets correlated to the NSCLC FlatIron biomarker data distribution. We then compute the average of the multiply imputed datasets and the spread of the *S* values. We will use S=200, a number seeking to support uncertainty due to few samples and long computational time.

#### 2.3.5. Imputation Models

Imputation models included the five biomarkers status EGFR, ALK, KRAS, PDL1, BRAF. Additionally, we use the target survival as a predictor variable as suggested before [9] to improve the data efficiency of the imputation model. The marginal Nelson-Aalen cumulative hazard estimate (H) was perfectly correlated with time at risk (T); see Figure 4. Hence, we used H to improve the imputation model. The correlation among the biomarkers ranges from 0 to 0.2. To be consistent with all imputation methods, we used the same multivariate Cox proportional hazards model in all cases; see Equation (2).

#### 2.3.6. Synthetic Data Generation

To further evaluate the impact of missingness on the imputation models performance, we use structural causal models to generate synthetic data that follows a specified MAR mechanism. The following running example is a standard structural causal model with real-world applications [27] used here for demonstration. Let *A* and *B* be latent variables, i.e., not observed. *A* causes *B*. *A* also causes three variables $A_{1},A_{2},A_{3}$ and partially causes $B_{1}$. *B* causes the manifest variables $B_{1},B_{2},B_{3}$. The variables $A_{1},A_{2},A_{3},B_{1},B_{2},B_{3}$ are manifest, i.e., observable. We generate sample datasets with: (9)$$\begin{array}{c} \begin{array}{c} B≔f_{B}\left(A\;\right)\\ A_{1},A_{2},A_{3}≔f\left(A,U_{A_{1},A_{2},A_{3}}\right)\\ B_{1},B_{2},B_{3}≔f\left(A,B,U_{B_{1},B_{2},B_{3}}\right) \end{array} \end{array}$$

Notably, $A_{1},A_{2},A_{3},B_{1},B_{2},B_{3}$ have random noise attached. From this model, we adjust the linear coefficients to obtain two different datasets, see Figure 5, with different levels of correlation between the manifest variables, such that:dataset type I has a high correlation ρ≈0.8.dataset type II has a low correlation ρ≈0.2.

To understand the impact of sample size on the imputation accuracy of the compared algorithms, we generate various datasets with increasing sample sizes. We generate 200 datasets of each type and proceed to “amputate” values. For MAR, we use a multivariate missingness simulation method based on a multivariate amputation algorithm [12]. Multivariate amputation’s general idea is to define the probability that the nth individual’s ith variable is missing conditional on the nth individual’s other variables’ missingness or observed value such that:(10)$$P_{i}^{m}=\frac{p^{m}\left(i\right)\cdot N\cdot \exp\left(-\sum_{j\neq i}w_{j}m_{j}\left(n\right)x_{j}\left(n\right)\right)}{\sum_{l=1}^{N}\exp\left(-\sum_{j\neq i}w_{j}m_{j}\left(l\right)x_{j}\left(l\right)\right)}$$

$p^{m}\left(i\right)$ corresponds to the proportion of missingness, and the $w_{j}$ are pre-specified weights.

Our approach to simulate synthetic survival datasets is to define the time-to-event process via a full proportional hazard model with all features $A_{1},A_{2},A_{3},B_{1},B_{2},B_{3}$ in the model and simulate survival time via inverse transform sampling see Algorithm A2. To be consistent with all imputation methods, we used the same full multivariate Cox proportional hazards model, with all features $A_{1},A_{2},A_{3},B_{1},B_{2},B_{3}$, as predictors.

## 3. Results

### 3.1. Synthetic Data Experiments

Using Equations (Equation 9) we experiment with increasing sample size: 625, 1250, 2500, 5000 and 10,000. Figure 6 shows the RMSE for each imputation algorithms. In the high correlation setting (ρ≈0.8), every method performs better than in the low correlation setting (ρ≈0.2). The trend implies that MITABNET and GAIN consistently outperform each benchmark across sample size and correlation settings.

We now compare the same benchmarks considering the bias and coverage in estimating the log-hazard ratio for the synthetic data. The log-hazard ratio $\bar{\beta }$ after imputation should be as near to the original $\hat{\beta }$ as possible, and the confidence interval 95% CI should overlap with the truth $\hat{\beta }$. Table 3 shows the results of percentage bias and Table 4 shows the results of coverage for low (ρ=0.2) and high (ρ=0.8) correlated scenarios and a sample size of 10,000.

### 3.2. Real-World Data Experiments

In our real-world data experiment, we assess the bias and coverage of parameter estimates and the impact of missingness for each imputation algorithm, see METHODS section. We also analyse the interval width. For methods that rely on the Gibb sampler, such as PMM and MITABNET, the convergence of the methods was visually checked and evaluated, see Figure A1, and $\hat{R}$ was near one, $0.9\leq \hat{R}\leq 1.1$, for each imputed feature.

Each imputation method’s percentage bias and coverage was assessed with 200 Flatiron NSCLC biomarker amputated dataset draws, using Algorithm A1. Then, multiple imputations were performed with each imputation method, obtaining five imputed datasets. Rubin’s rules, see Equations (Equation 3) and (Equation 7) were used to obtain 200 imputed parameter estimates $\bar{\beta }$ and 95% CI for each of the six methods.

Table 5 shows the percentage bias of each method, which depends on the missingness and the concept drift. For instance, the on-diagonal elements of Table 5 indicate the impact of missingness for each imputation algorithm. The off-diagonal elements indicate the additional difficulty for the algorithm to learn the concept drift. The model is re-imputing a dataset that was imputed originally with another algorithm.

Table 6 shows the coverage of the 95% CI. Similarly to the percentage bias experiment, the on-diagonal elements of Table 6 indicate the impact of missingness on coverage and the off-diagonal the impact of concept drift.

As we can see in Table 7, the interval width for each imputation algorithm depends more on the concept drift than on the missingness. In general, an algorithm is best if it has a smaller confidence interval width with higher coverage. Therefore, Table 7 interpretations need to consider together Table 6, which showed coverage results. Finally, Appendix D depicts the ground truth parameter estimates beta, regarding the different concept drifts and the parameter estimates considering each algorithm.

## 4. Discussion

Several methods have recently been proposed to perform multiple imputations with missing data for RWE observational datasets [10,11]. To our knowledge, few papers have systematically compared the statistical properties of the various methods, considering the impact of missing data and the concept drift. To help potential research on RWE datasets choose an imputation method, we have studied six methods that perform multiple imputations.

All algorithms draw multiple imputations using different but comparable techniques. PMM and MITABNET use a Gibbs sampler approach; MIRF uses different random seeds to initialise a random forest; EM and MIPCA use bootstrap-based approach; GAIN uses generative adversarial networks. A Deep Neural Network powers both GAIN and MITABNET, where multiple imputations may be drawn by applying dropout layers [28] at training and predicting imputations time. To our knowledge, we are the first to investigate the usefulness of Tabnet as an algorithm for multiple imputations (MITABNET) and systematically compared it with state-of-the-art methods. MITABNET can become part of the pre-processing step of covariates for RWE dataset analysis, combining the interpretation of multiply imputed datasets for more robust inference.

In this article, we have focused on finding the best imputation method for realistically complex datasets. Our synthetic data experiment used a structural causal model to sample multivariate datasets with different levels of correlation among the observed features. As seen in Figure 6 and Table 3, all methods perform best when the correlation among variables is high. Our results agree with previous research showing that the best-case setting for applying off-the-shelve imputation algorithms is the MAR mechanism with a high correlation between variables. The synthetic data experiment found that MITABNET and GAIN outperformed every other algorithm in high and low correlation settings, using accuracy and percentage bias. However, analysing the RWE NSCLC FlatIron dataset did not find conclusive results of the best method considering missingness impact and concept drift. Only three methods, MIRF, MITABNET and PMM, achieved low percentage bias for the scenario where the concept drift was in favour of them, also showing low percentage bias for the impact of missing data <20%. As seen in Table 5, PMM achieved consistently acceptable coverage >50%, only outperformed under the concept drift of MIRF and MITABNET. On the other hand, PMM also had the most extensive confidence intervals across all imputation algorithms.

We analysed the bias and coverage of parameter estimates after imputing with several imputation algorithms, extending the approach for a standardised evaluation of imputation algorithms from [10,11], which concluded that MIRF or GAIN result in more accurate imputation and sharper inference than other imputation algorithms. Our synthetic data results indicate that tabular networks may outperform randomised decision trees and generative adversarial networks for low and high correlated datasets with the structural causal model used previously by [27], being less biased and hence, preferred for sharper inference.

## 5. Limitations

Although we performed the present comparative study with realistically complex analyses and real-world data, it has limitations. The most critical limitation is that our results are dataset-dependent. A limitation of our imputation algorithm is that to avoid an excessive computational burden, we only performed five multiple imputed datasets for each method, leading to potentially noisy between-imputation variability. For realistic analysis, [25] recommended estimating the number of imputations necessary to produce efficient estimates by conducting a relative efficient analysis of the fraction of missing information [29]. Nevertheless, the default choice for the most popular multiple imputations packages is five [8], and although we evaluated the convergence of the algorithms, it is possible that analysing RWE datasets need more imputations to produce efficient estimates.

Finally, our study focused on MAR missingness patterns. However, an MNAR missing data pattern may be unknown in practice, and results should be generalised with caution. Alternatives to pre-canned algorithms, such as full information maximum likelihood [30], and full Bayesian imputation [5], where the missing values’ model assumptions are explicit in the model formulation, may be more appropriate for MNAR settings. However, full information maximum likelihood and fully Bayesian approaches require extra engineering steps to include the missing variables in the model and are out of the scope of this analysis. Algorithms for multiple imputations such as MITABNET work well for MAR and remain the standard approach for handling missing data with imputation algorithms [8,31].

## 6. Conclusions

The multiple imputations approach is the standard approach for handling missing data in RWE datasets, and we have shown a new method to compare algorithms that perform multiple imputations. MITABNET is a promising algorithm to draw multiple imputations for complex datasets with non-linearities. MITABNET appears to outperform conventional methods such as multiple imputations with PMM for RWE datasets examples, such as the FlatIron NSCLC biomarker dataset.

## Figures and Tables

**Figure 1 jpm-11-01356-f001:**
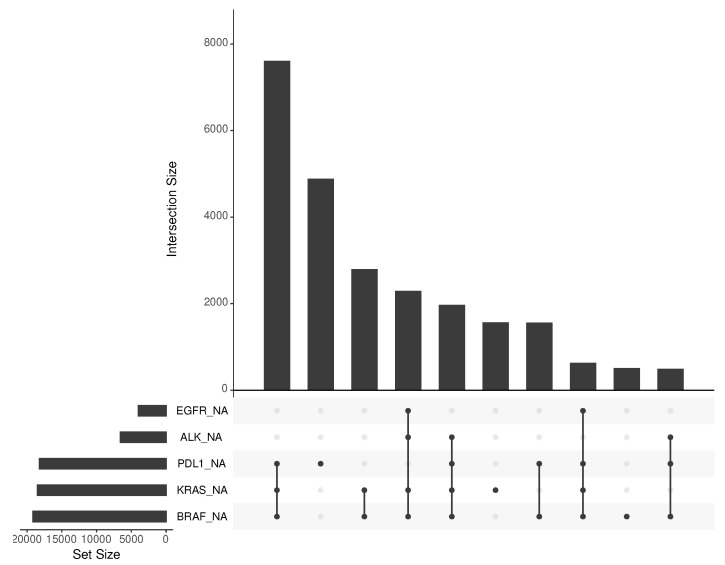
Biomarker status missingness and its combinations of missingness using the intersecting sets visualisation method [15].

**Figure 2 jpm-11-01356-f002:**
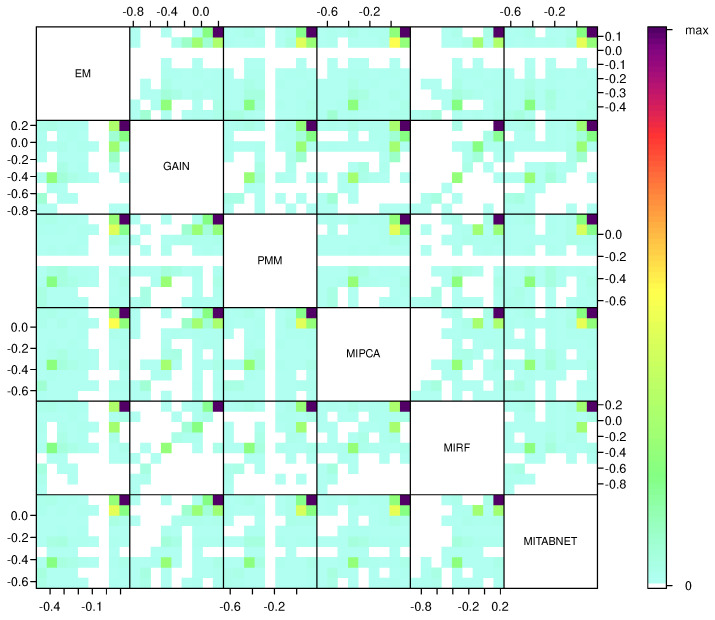
Scatter plot matrix of prognostic indexes (μ) and pairwise comparisons using the six different imputation methods on the off-diagonal for the first random imputation sample of the Flatiron NSCLC analytical cohort. We use the visualisation method for the image scatter plot matrix of large datasets using pixel density from [23].

**Figure 3 jpm-11-01356-f003:**
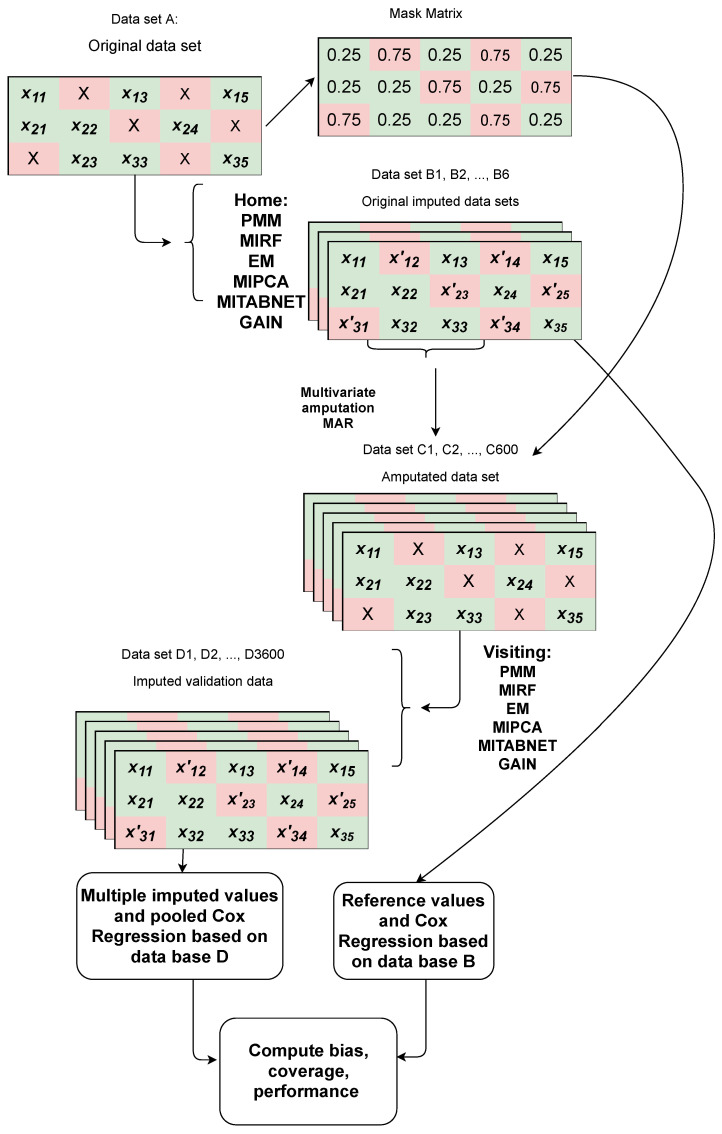
Generation of datasets with artificial missingness from a population of patients with NSCLC in the Flatiron database. datasets B1, B2, ..., B6 are imputed datasets with the imputation algorithms (PMM, MIRF, EM, MIPCA, MITABNET, GAIN) serving as host to the comparison or imputing at home. datasets C1, C2, ..., C600 are amputated dataset 100 for each B dataset. datasets D1, D2, ..., D3600 are imputed datasets 6 for each C dataset, the imputation algorithms are visiting.

**Figure 4 jpm-11-01356-f004:**
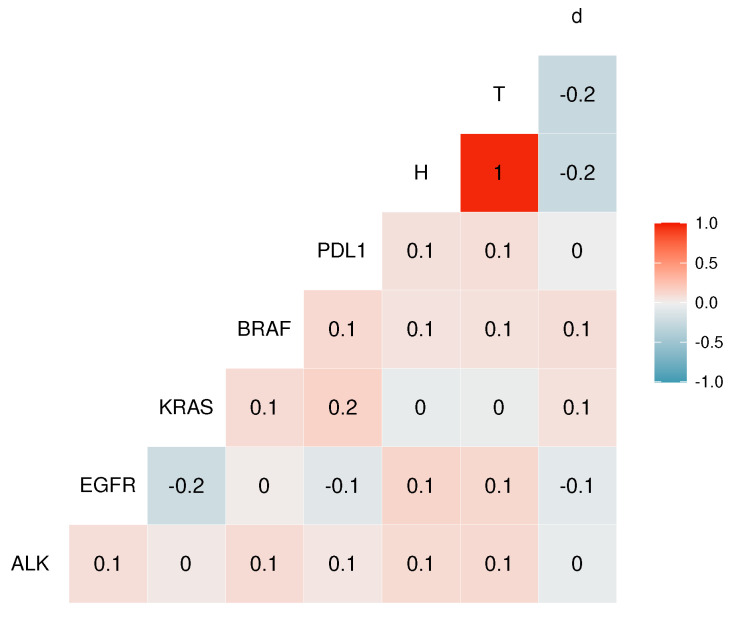
Pearson correlations between the biomarker status ALK, BRAF, EGFR, KRAS, PDL1, cumulative death hazard H, survival time T, and death status in the Flatiron NSCLC dataset.

**Figure 5 jpm-11-01356-f005:**
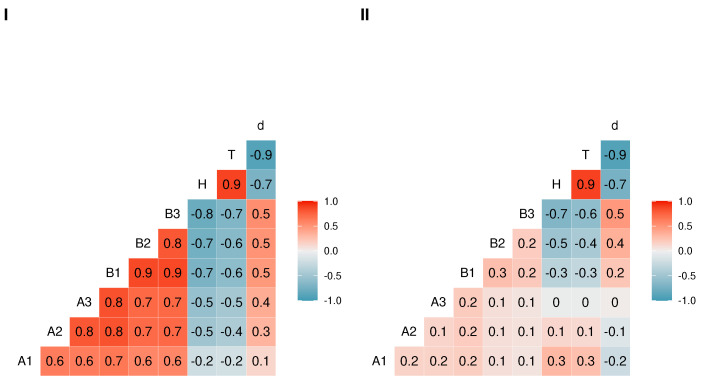
Pearson correlations between the features $A_{1},A_{2},A_{3},B_{1},B_{2},B_{3}$, cumulative death hazard H, survival time T, and death status in synthetic datasets type (**I**) (high correlation) and (**II**) (low correlation).

**Figure 6 jpm-11-01356-f006:**
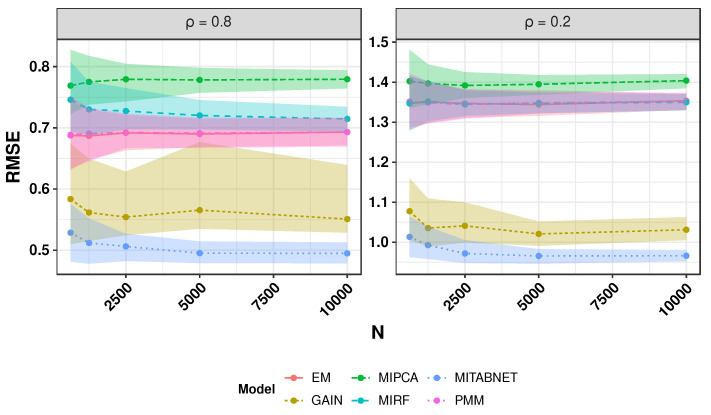
Head-to-head comparison of imputation algorithms in post-imputation accuracy in high and low correlation scenario and increasing sample sizes.

**Table 1 jpm-11-01356-t001:** Status of biomarkers for the population cohort of NSCLC patients followed-up for the present study.

Biomarker	Positive	Negative	Unknown
ALK	971 (2.77%)	27,612 (78.86%)	6429 (18.36%)
EGFR	5196 (14.84%)	25,928 (74.05%)	3888 (11.10%)
KRAS	4778 (13.65%)	12,247 (34.98%)	17,987 (51.37%)
BRAF	847 (2.42%)	15,579 (44.50%)	18,586 (53.08%)
PDL1	6052 (17.29%)	11,301 (32.28%)	17,659 (50.44%)

**Table 2 jpm-11-01356-t002:** Values of the percentage bias for three imputation methods using two imputed bootstraps from the NSCLC Flatiron dataset.

Method	Sample 1			Sample 2		
	EM	PMM	MITABNET	EM	PMM	MITABNET
EM	34.7	**18.6**	54	27	**19.2**	46.8
GAIN	53.6	61.6	**8.5**	57.4	69	**4.9**
PMM	90.5	**20.3**	99	79.3	**7.8**	75.8
MIPCA	64.9	**8.5**	66.1	61.1	**14**	83
MIRF	117.3	**24.4**	119.2	96.1	**15.3**	133
MITABNET	**6**	24.5	13.4	6.9	28.4	**2.2**

**Table 3 jpm-11-01356-t003:** Percentage bias in synthetic data: high and low correlation scenario for a sample size of 10,000.

	Percentage Bias	
Model	ρ = 0.8	ρ = 0.2
EM	11.5±0.6	74.9±1.5
GAIN	10.5±1.3	41.5±1.7
PMM	10.1±1.2	74.8±1.7
MIPCA	15.7±1.3	95.2±1.7
MIRF	10.8±0.7	74.3±1.7
MITABNET	2.6±1.4	36.2±1.4

**Table 4 jpm-11-01356-t004:** Coverage in synthetic data: high and low correlation scenario for a sample size of 10,000.

	Coverage	
Model	ρ = 0.8	ρ = 0.2
EM	0.42	0.12
GAIN	0.25	0.07
PMM	0.43	0.13
MIPCA	0.20	0.05
MIRF	0.39	0.11
MITABNET	0.75	0.14

**Table 5 jpm-11-01356-t005:** Percentage bias for each imputation algorithm in the Flatiron NSCLC dataset.

Home	EM	GAIN	MIPCA	MIRF	MITABNET	PMM
EM	33.9 ± 5.8	121.5 ± 111	13.5±3.6	28.6 ± 13.6	47.6 ± 4	16.7 ± 3.4
GAIN	53.1±4.1	81.9 ± 36	69.3 ± 2.4	66.6 ± 3.4	95.8 ± 90.6	66 ± 5
MIPCA	62.8 ± 7.2	123.3 ± 116.6	26.8 ± 8.1	8.4±7.8	71.1 ± 16.2	9.2 ± 4.6
MIRF	106.8 ± 7.1	100.8 ± 87.8	55.2 ± 5.5	9.1±3.4	130 ± 15.5	25.8 ± 5.6
MITABNET	3.8±3.3	80 ± 74.1	4.3 ± 3.7	57 ± 3.9	9.8 ± 5.7	27.5 ± 4.9
PMM	85.9 ± 8.2	52.5 ± 43.5	47.1 ± 11.7	11 ± 9.6	102.3 ± 19.5	10.8±7

**Table 6 jpm-11-01356-t006:** Coverage for each imputation algorithm in the Flatiron NSCLC dataset.

Home	EM	GAIN	MIPCA	MIRF	MITABNET	PMM
EM	0.50	0.38	0.72	0.56	0.52	0.68
GAIN	0.18	0.22	0.32	0.34	0.14	0.50
MIPCA	0.36	0.40	0.50	0.90	0.40	0.94
MIRF	0.32	0.20	0.40	0.68	0.22	0.62
MITABNET	0.68	0.12	0.62	0.32	0.88	0.70
PMM	0.22	0.22	0.10	0.32	0.24	0.90

**Table 7 jpm-11-01356-t007:** Interval width for each imputation algorithm in the Flatiron NSCLC dataset.

Home	EM	GAIN	MIPCA	MIRF	MITABNET	PMM
EM	0.09	0.11	0.11	0.13	0.10	0.19
GAIN	0.11	0.11	0.12	0.12	0.13	0.16
MIPCA	0.10	0.12	0.12	0.14	0.11	0.30
MIRF	0.12	0.12	0.14	0.15	0.12	0.20
MITABNET	0.08	0.08	0.09	0.09	0.10	0.16
PMM	0.11	0.12	0.12	0.15	0.12	0.28

## Data Availability

Synthetic data and code to reproduce the synthetic data experiment are available at https://github.com/csetraynor/miml-synthetic. FlatIron NSCLC data is available from FlatIorn Health, and the code for the real-world data experiment is available from the authors under request.

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
