# Peer review of "Imputing Biomarker Status from RWE Datasets—A Comparative Study"

_jpm, 2021, doi:10.3390/jpm11121356_

Round 1
Reviewer 1 Report
The authors makes significant contributions to deal with the universal challenge of working with missing values and RWE datasets, with robust methods and results.
Author Response
Dear Reviewer,
The authors thank your comments and, following your suggestions, have made the following improvements:
- The spelling has been reviewed and fine improvements made.
Reviewer 2 Report
Dear author
the paper resulted very interesting. I have minor comments.
the description of performace (lines 259) should be moved before the use of percentage bias at line 236 or you should find a better union among lines.
In my opinion the lines 286-302 are still methods, I invite you to evaluate how to shorten this part in the results chapter or to move into methods.
Author Response
Dear Reviewer,
The authors thank your comments and, following your suggestions, have made the following improvements:
- The Methods section now defines percentage bias before being used as an example in the "multiple comparisons" subsection.
- The Methods section now includes a "synthetic data generation" sub-section to define further the process for generating synthetic data to evaluate the impact of missingness in the imputation methods. The results section is now shorter and focuses only on the results as suggested.
I highlight the changes in the attach.
